# Spatiotemporal pattern of land use and land cover changes of upper Sangu-Matamuhuri Watershed in the South-Eastern Bangladesh

Ajit Kumar Rudra◉, A. K. M. Rashidul Alam◉◉*

Department of Environmental Sciences, Jahangirnagar University, Dhaka, Bangladesh

◉ These authors contributed equally to this work.
* rashidul@juniv.edu

## Abstract

This study aims to analyze the spatiotemporal patterns of land use and land cover (LULC) changes in the upper Sangu-Matamuhuri Watershed (SMW) of Bangladesh from 1988 to 2019 using integrated remote sensing (RS) and geographic information system (GIS) approaches. The watershed was divided into five major land use land cover classes namely forest, agriculture, shrubland, settlement, and water body. From the analysis it was observed that the forest cover showed a major variation with the interval, decreasing from 234634 ha (76%) in 1988 to 168476 ha (54%) in 2019 with an annual loss of 2134.13 ha/year. The temporal overlay operation revealed that forest patches were converted to shrubland due to deforestation and forest burning. This shrubland was then cleared or re-burned for shifting cultivation and significantly increasing agricultural land from 21787 ha (7%) in 1988–78704 ha (25%) in 2019 at the rate of 1836.03 ha/year. The area of other land use types including shrubland and settlement also increased. However, the area of water body, the only source of drinking water to the indigenous community and irrigation water in the dry season, was found decreased with an average annual rate of 29.84 ha/year which indicate the drying of water body. These LULC changes not only threaten watershed resources and ecosystem services but also pose significant risks to local communities' dependent on these resources for livelihood and water supply. Without proper management of watershed, these resources will soon be lost and no longer be able to play their role in socioeconomic and environmental development of the area. The findings of this study have important policy implications for sustainable watershed management in the study area as well as in other similar hilly regions of the tropics.

## Introduction

Forests play a vital role in preserving wildlife, generating wood and timber, and balancing the micro-climate, local economy, agro-ecology and subsurface biota.

**Data availability statement:** All relevant data are within the manuscript and its Supporting Information files.

**Funding:** The author(s) received no specific funding for this work.

**Competing interests:** The authors have declared that no competing interests exist.

However, as the population increased, these natural resources have been utilized abruptly in order to serve their increasing demand in different parts of the world [1]. The lack of knowledge and understanding in relation to the environment and functioning of ecological systems led to the unsustainable exploitation of natural resources and fewer opportunities for resource utilization [1–3]. Changes observed in the biophysical cover on the earth's surface (including vegetation, grasslands, or water bodies) due to anthropogenic utilization for construction or conservation purposes are often termed land use and land cover (LULC) changes [4–5]. Research on LULC changes using remote sensing technology has a long history and has advanced [6–12]. This is due to the availability of air-borne and space-borne remote sensing platforms and sensors that enable the observation of biophysical attributes over large areas at various spatial, spectral, and temporal scales [13]. Changes in LULC are considered an important indicator in understanding the interaction between human activity and the environment [11]. Change detection studies concentrate on identifying the biotic and abiotic components of the spectral and temporal changes that take place within watersheds [14]. To assess the variation in LULC using satellite data, a variety of change detection methods have been developed and used such as image differencing, vegetation index differencing, principal component analysis, and post-classification comparison [6,15]. Spectral mixture analysis, artificial neural networks, and integration of geographical information systems and remote sensing data have recently emerged as essential techniques for change detection applications. Different change detection algorithms have their own advantages and no single approach is optimal and applicable for all cases. Spatial change analysis has increasingly been recognized as one of the most important tools for effective watershed management [16–18]. Because it determines the hydrological and ecological processes taking place in a watershed [19,20]. Satellite remote sensing coupled with GIS techniques has been widely applied in spatial change analysis [21–23]. In this technique, GIS layers are directly overlaid on image data and the image processing results are moved into a GIS system to generate LULC changes, including information, such as the trend, rate, nature, location, and magnitude of the changes [8,15]. Thus, spatiotemporal mapping and proper LULC change analysis are considered essential for the sustainable watershed management.

In developing countries like Bangladesh extensive areas have been undergoing LULC changes since long ago [24]. Afforestation and deforestation activities often result in substantial changes in terms of land area and hydrological impacts [16,25]. Several studies relate population growth and deforestation in developing countries [2,26,27]. Unsustainable land use practices including swidden agriculture, deforestation, overcutting of fuel wood and timber, excessive grazing, improper water use, and construction of roads along with other factors have been putting increasing pressure on hill forest areas of many countries including Bangladesh [3,28]. As a result, watershed degradation occurred in these areas at various temporal and spatial scales [28] resulting in high-intensity storms causing flash floods, landslides, declining soil productivity, unfertile and erosive soil having limited depth, and moisture holding

capacity, sedimentation in stream channels and polluted water [29–30] which ultimately affecting quality of life by causing food deficit, low farm income, health hazards, etc. among others [3,31].

Sangu-Matamuhuri Watershed (SMW) is located in the south-eastern part (Bandarban District) of Bangladesh which is not an exception to the above-stated problems [32]. Sangu and Matamuhuri, the two vibrant rivers of Bandarban district, are now slowly deteriorating due to climate variability, causing a navigability crisis and serious water scarcity. Water levels have decreased as a result of shifting cultivation, deforestation, hill-leveling, unrestricted stone extraction, and unplanned water extraction for agricultural and household uses [33]. Silt accumulation has hampered navigation and led to the extinction of many fish species. More than 80% of the Bandarban population, who depend on the rivers for water, now face acute water shortages due to the drying of the rivers [34]. In the south-eastern tertiary hilly areas of Bangladesh, significant reductions in hill forests and water reservoirs have been noted with continuous increases in shrubland, agricultural land, and settlements [35–36]. Reddy et al. revealed that forests cover in Bangladesh had been significantly reduced in terms of area and quality. The huge loss of forest cover was found from 1930 to 2014 when most forest conversions resulted in the degradation of forests to scrub and transition to agriculture and plantations [37]. In view of the steady and rapid decline of natural resources and the accompanied environmental problems, it is essential to properly monitor and analyze the dynamics of these LULC changes.

A few studies have so far been carried out to explore the management of agricultural systems for sustainable food security [38]; biological diversity [39]; stream flow characteristics [3], socio-economic determinants affecting landscape restoration [40], land use and land cover change of Chittagong Hill Tracts (CHTs) in general [35–36]. But little is known about LULC change dynamics of Sangu-Matamuhuri watershed.

Despite these advancements in LULC research, there remains a lack of comprehensive spatiotemporal analysis of LULC changes in the Sangu-Matamuhuri Watershed. Therefore, this study aims to carry out intensive analysis depending on long term data on LULC change dynamics in the upper part of SMW on both spatial and temporal basis using GIS-RS technology which could help to identify the limiting factors in the study area and thereby help policy makers to adapt suitable conservation measures. The specific objectives of this study are: i) to develop LULC maps of the study area for the years 1988, 1999, 2009, and 2019; and ii) to analyze how the different LULC of the study area changed from one to another, i.e., forest to agriculture, settlement, shrubland, etc. over the period of time (1988–2019) in order to derive insights into land use transitions and so as to help the policy maker to adapt suitable conservation measures.

## Materials and methods

### Study area

SMW is located in the southeastern part of Bangladesh stretching throughout Chittagong and Cox'sbazar districts in the lower part and Bandarban district of CHT's in the upper part. The study area was delineated using satellite imagery with watershed boundary generated from ASTER (Advanced Space borne Thermal Emission and Reflection Radiometer) DEM (Digital Elevation Model) using the Arc hydro tool in ArcGIS 10.5 environment [3]. The study area, as shown in Fig 1, includes the upper part of the Sangu-Matamuhuri Watershed in the southeastern region of Bangladesh. It covers an area of about 3,10,317 ha and extends between 21° 14' 22.249" N to 22° 19' 13.983" N latitude and 92° 11' 41.561" E to 92° 40' 41.676" E longitude with an elevation ranges from 0 m to 1027 m. Annual rainfall in this area ranges from 2400–3000 mm.

The main land use is swidden agriculture (locally known as Jhum), a slash-and-burn farming method. Cultivating multiple agricultural crops in the cleared patch of hill slopes for one or two seasons and then shifting to another place is a major trait of this land use. Several studies revealed that loss of soil and nutrient and sediment deposits were consequences of this agricultural practices at steep and very steep slopes of hills and ridges [38,41].

Fruits including Banana (*Musa sp.),* Pineapple (*Ananus comosus*), Jackfruit (*Artocarpus heterophyllus*), and Papaya (*Carica papya*), undergrowth including Ginger (*Zingiber officinale)*, Turmeric (*Curcuma longa*) and tribal textiles are the

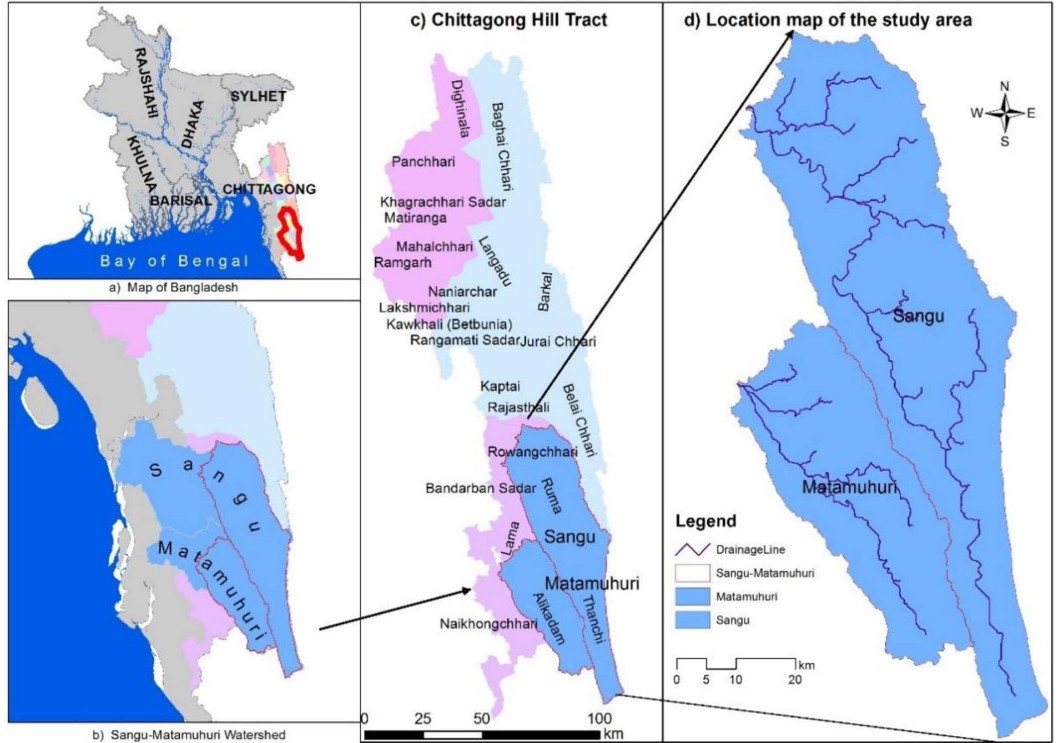

**Fig 1. Study area of SMW showing a) Map of Bangladesh, b) SMW area in full extent, c) CHT's indicating study area, d) Study area with its main stream network.**

main exportable items of the region. Tourism due to presence of many tourist attraction spots including natural lake, water spring, beautiful hilly landscape with green cover, indigenous peoples culture and also the flowing river serve as a growing source of revenue. Bandarban district has the largest portion of jhum land (57%) followed by two other adjacent hill districts of CHTs including Rangamati (32%) and Khagrachhari (11%) [36,40,41].

A survey was conducted at seven different sites to diagnose farming practices and their impacts on soil resource loss in the CHTs [36]. The survey found that the loss of soil and nutrient and sediment deposits were consequences of agricultural practices and operations at steep and very steep slopes of hills and ridges.

Population estimates for the watershed area, according to the census conducted in 1981 and 2021, indicate that over a period of 40 years, the population in the area has gone up from 144,071 in 1981–321,464 in 2021, marking an increase of 123.12 percent as shown in Fig 2 [42,43]. Considering the same growth rate per year, the population of the area would be 485,035 in 2031. The increasing rate of population made necessary expansion of agricultural activities in the hilly areas at the expense of the environmental values of the watersheds [44].

## Data acquisition and image preprocessing

For this study, four Landsat 5 TM (Thematic Mapper) images, two Landsat 7 ETM (Enhanced Thematic Mapper) images, and two Landsat 8 OLI_TIRS (Operational Land Imager and Thermal Infrared Sensor) images with 30 m spatial resolution pertaining to the years 1988, 1999, 2009 and 2019 were selected for land cover mapping which were downloaded free of charge from United States Geological Survey (USGS) earth explorer (http://earthexplorer. usgs.gov/). The Landsat images were chosen depending upon data availability and decadal change monitoring purposes. Characteristics for each image are shown in Table 1.

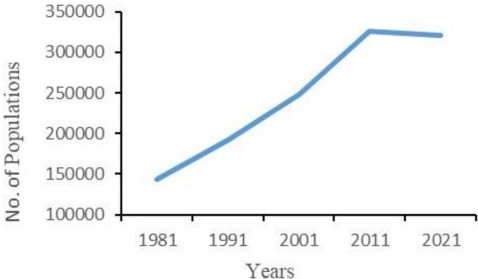

**Fig 2. Demographics of Sangu-Matamuhuri watershed area from 1981 to 2021.**

**Table 1. Image characteristics of Sangu-Matamuhuri watershed in the years 1988, 1999, 2009 and 2019.**

| Acquisition Date | Satellite (Sensor) | Path/Row | Resolution (m) | Cloud Cover (%) | Staked band and wave length |
|---|---|---|---|---|---|
| 02/21/1988 | Landsat 5 (TM) | 135/045 | 30 × 30 | 3 | Band 1: 0.45–0.52 |
| 02/12/1988 | Landsat 5 (TM) | 136/045 | 30 × 30 | 6 | Band 2: 0.52–0.60 |
| 12/19/1999 | Landsat 7 (ETM+) | 136/045 | 30 × 30 | 0 | Band 3: 0.63–0.69 |
| 02/14/2000 | Landsat 7 (ETM+) | 135/045 | 30 × 30 | 0 | Band 4: 0.76–0.90 |
| 01/13/2009 | Landsat 5 (TM) | 135/045 | 30 × 30 | 1 | Band 5: 1.55–1.75 |
| 01/04/2009 | Landsat 5 (TM) | 136/045 | 30 × 30 | 1 | Band 7: 2.08–2.35 |
| 01/25/2019 | Landsat 8 (OLI_TIRS) | 135/045 | 30 × 30 | 0.03 | Band 2: 0.452–0.512 |
| 02/01/2019 | Landsat 8 (OLI_TIRS) | 136/045 | 30 × 30 | 0.02 | Band 3: 0.533–0.590 |
| | | | | | Band 4: 0.636–0.673 |
| | | | | | Band 5: 0.851–0.879 |
| | | | | | Band 6: 1.566–1.651 |
| | | | | | Band 7: 2.107–2.294 |

All the images belong to the month of December, January, and February. The data set of 1999 was found hazy which might reduce the accuracy level. So, February 2000 image was considered, instead of 1999 [45] and others were chosen within the ranges of minimum cloud cover percentages for the consecutive year's imagery. UTM (Universal Transverse Mercator) coordinate system with zone 46 north and datum WGS 84 (World Geodetic System) was used for satellite imagery registration. The methodological workflow (Fig 3) outlines the steps from data acquisition and preprocessing to classification and accuracy assessment.

All bands except panchromatic and thermal bands from TM, ETM+, and OLI_TIRS were considered in this study for making multispectral imagery. Mosaicking of paths 135–136 and row 045 was carried out in order to get the full extent of the study area. For reducing classification error and increasing the accuracy of the staked imagery, atmospheric correction was performed using ATCOR 2 platform of ERDAS Imagine 2014.

The radiometric correction was carried out by converting the pixel values of each image to radiance values using sensor calibration equation (i) [46–47]. To avoid anomalies between two different tiles, color correction-based image dodging, and geometry-based weighted seam line polygon were incorporated which provide sufficient accurate information within the study area [48–49].

$$LTOA = \left( \frac{Lmax - Lmin}{QCAL\,max - QCAL\,min} \right) \times (DN - QCA\,Lmin) + Lmin$$

(i)

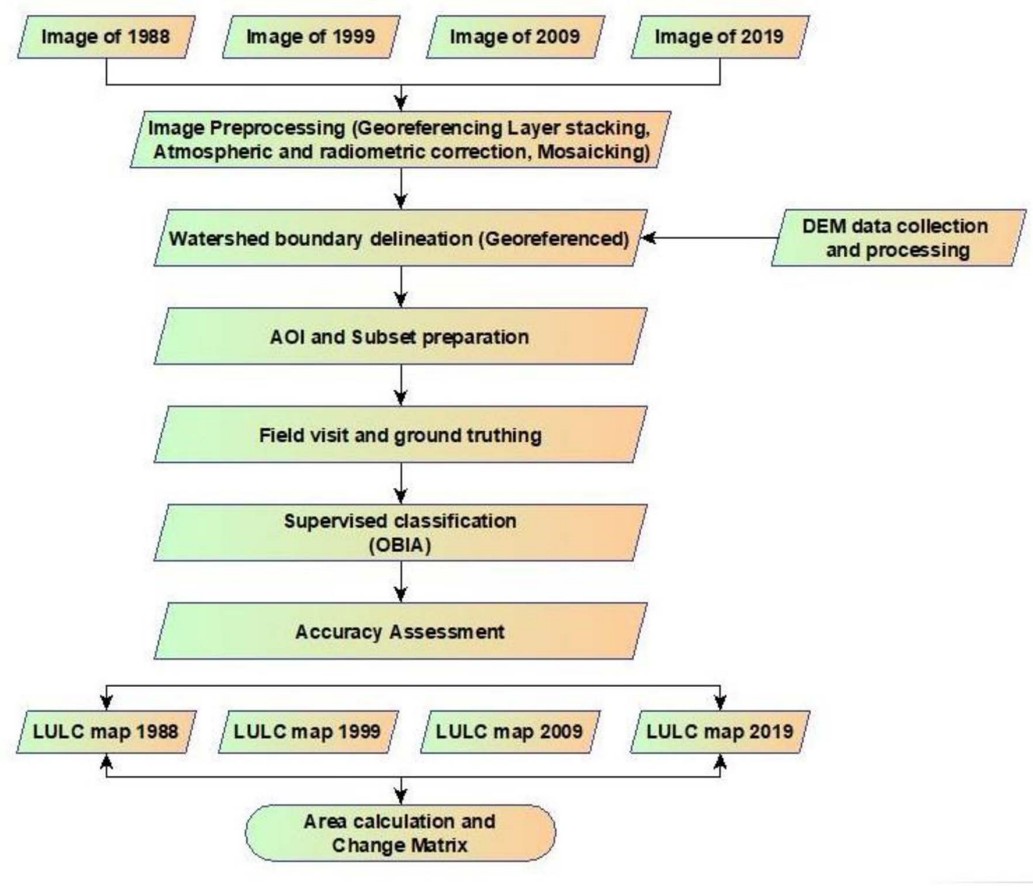

**Fig 3. Methodological workflow used in the study.**

where $L_{TOA}$ is solar radiance at the top of the atmosphere, $L_{max}$ and $L_{min}$ are maximum and minimum radiance in W/m$^{-2}$ sr$^{-1}$ μm$^{-1}$. QCAL$_{max}$ and QCAL$_{min}$ are maximum and minimum DN values possible (255 or 1). The calibration gains and offset ($L_{max}$ and $L_{min}$) are available in the imagery header file.

## Field sampling and classification algorithm

Imagery registration with field-level real features including texture and color is required for signature generation and further classification work which has been done for this study in November 2017 [50]. Handheld GPS Garmin eTrex78, mobile devices, and a camera for geotag photography were used for registering each GPS location. Landsat-8 scene of 2019 (path 135/136, row 45) was considered as the reference image for registering others. GPS locations were dispersed throughout the images, to make sure that the root mean square (RMS) error was less than 0.5 pixels. A field survey map along with ancillary datasheets was prepared before the field visit. For image registering and cross-matching purposes, 70 final points among 96 GPS points were used, rest points along with random sampling points were used for sampling and accuracy assessment of land use and land cover (LULC) classes.

Supervised image classification by user-defined pixel or sample is the fundamental step for LULC change analysis [51] and visual interpretation [52]. Object-based image classification is a more accurate method [52] and advantageous than pixel-based image classification [53,54]. An object-based image classification approach was followed for this study which involves a number

of steps like multispectral segmentation with scale parameter 10, shape 0.1, and compactness 0.5. User-defined known segments or samples were selected for identifying unknown segments based on reflectance, color, shape, and pattern and standard Training and Test Area (TTA) masking was used for appropriate covariant sample selection purposes [55]. In order to derive various LULC classes, a classification scheme was developed based on the prior knowledge of the study area for over 20 years and a brief reconnaissance survey with additional information from previous research [56–58] in the study area. Five LULC classes were identified such as 1) Forest, 2) Agriculture, 3) Shrub land, 4) Settlement, and 5) Waterbody as shown in Table 2 [57]. For every class minimum of 50 samples (segments) were defined and nearing neighborhood classification was used following similar research work [59]. In this study nearing neighborhood method with applying standard neural network (NN) value to classes was used for LULC classification. On screen digitizing was performed for correction of misclassified segments.

## Accuracy assessment

In order to find the reliability and accuracy of the maps produced, an accuracy assessment was performed by developing an error matrix sometimes called confusion matrix which shows the relationship between ground truth data and the corresponding classified data [60]. Corresponding producer's accuracy, user's accuracy, and overall accuracy [61–63] and kappa coefficient (K) [64] were calculated by using equations ii, iii, iv and v, respectively. Kappa value above 0.80 is generally considered very good [61–62]. Real-world 30 GPS points (collected during field visit based upon accessibility and image coverage), google earth driven 2.5m Very High Resolution (VHR) imagery, and Bing mapping were used for estimating accuracy level.

$$Producer's\ accuracy = \frac{Number\ of\ points\ correctly\ classified}{Total\ number\ of\ validation\ points\ identified\ as\ a\ true\ class} \tag{ii}$$

$$User's\ accuracy = \frac{Number\ of\ ponts\ correctly\ classified}{Total\ number\ of\ validation\ points\ mapped\ as\ a\ true\ class} \tag{iii}$$

$$Overall\ accuracy = \frac{The\ numberof\ correct\ ponts\ or\ pixels\ that\ make\ the\ major\ diagonal}{The\ total\ number\ of\ points\ or\ examined\ pixels} \tag{iv}$$

$$Kappa\ co-efficient\ (K) = \frac{(Po-Pe)}{(1-Pe)} \tag{v}$$

where Po = Observed proportion of agreements ($\sum_{i=1}^{r} X_{ii}$)
Pe = Expected proportion of agreements ($\sum_{i=1}^{r} X_{i+} X_{+i}$)
r = number of rows and columns in error matrix,
N = total number of observations (pixels),
X ii = observation in row i and column i,
Xi+ = marginal total of row i,
X+i = marginal total of column i,

**Table 2. LULC classification scheme used for the study [57].**

| LULC category | Description |
|---|---|
| Forest | Natural forest disturbed or secondary growth forest and reforestation. Natural forests include evergreen forest (tropical wet evergreen, tropical mixed evergreen and bamboo forests) and open deciduous forest type. Reforestation includes Teak (*Tectona grandis*), Eucalyptus (*Eucaluptus camaldulensis*), Acacia (*Acacia auriculiformis, Acacia mingium*) and mixed forest plantation |
| Agriculture | Paddy field, shifting cultivation, upland rice, mixed orchard, perennial trees – mango, tamarind, jackfruit, pineapple and pasture, farm house (cattle, fish, and poultry) |
| Shrub land | Bushy areas comprising herbs/shrubs/scrub |
| Settlement | Lowland village, institutional land, district town, recreational areas with semi-settled zone and factory |
| Water | River, reservoir, lake, farm pond and canals |

## LULC change analysis

In order to investigate the changes in LULC of SMW over the period of 1988, 1999, 2009, and 2019, the areal distribution of the four classified images was first calculated by different LULC categories and then the trend of changes was analyzed. Magnitude change was calculated by subtracting the area coverage from the 2nd year and initial year (Eq. vi). Percentage change was then calculated by dividing magnitude change by the area of the base year and multiplied by 100 (Eq. vii and viii). Annual rate of change was calculated by dividing magnitude change by the number of study years as shown in Eq. viii [65].

$$Magnitude\ change\ = Area\ of\ 2nd\ year\ -\ Area\ of\ the\ base\ year \qquad \text{(vi)}$$

$$Percentage\ change = \frac{Magnitude\ change}{Area\ of\ the\ base\ year} * 100 \qquad \text{(vii)}$$

$$Annual\ rate\ of\ change = \frac{Magnitude\ change}{Number\ of\ the\ study\ years} \qquad \text{(viii)}$$

Classification results were summarized by using Arc Map 10.5 and other programs like MS Excel, pivot table feature, and so on to create a table of conversion or change matrix among LULC classes for the four periods. Change matrix presents important information about the spatial distribution of changes in LULC and critical insights into land use transitions including forest to agriculture, shrubland and settlement growth; shrubland to settlement and forest; agriculture to forest or settlement etc. which will help the policy maker to adapt suitable conservation measures [66]. In this study earlier and later images were overlaid to generate change matrices. Between the two periods, the loss and gain areas of each land use class were derived from the change matrices by subtracting the remaining area from its total area.

## Results

### Mapping Accuracy and LULC change status

An accuracy assessment of LULC classes for the years 1988, 1999, 2009, and 2019 is shown in Table 3. Across the study area, overall classification accuracy was found 92%, 93%, 94%, and 94% for 1988, 1999, 2009, and 2019, respectively with their corresponding k statistics of 0.90, 0.91, 0.93, and 0.93. User's and producer's accuracy values of individual classes were consistently high, ranging from 70% to 100% and from 81% to 100%, respectively. Accuracy results indicate that the LULC maps are acceptable enough and can be used for LULC change analyses and projections in the study area.

Fig 4 displays the multi-temporal LULC maps, illustrating the spatial distribution of forest, agriculture, shrubland, settlements, and water bodies across the years 1988, 1999, 2009, and 2019. Spatial distribution of each thematic class identified in the SMW for the years 1988, 1999, 2009, and 2019 respectively are presented in Fig 5 and Table 4. The magnitude and percentage changes were calculated using equations (Eq. vi and vii) as described in the methods section.

Results from classified maps indicated that in the year 1988 forest land formed a major part of the upper part of Sangu-Matamuhuri watershed i.e., 2,34,634 ha or 76% of the total study area. With the passage of time the forest cover showed a major variation as it became notably decreased to 1,97,852 ha (64%) in 1999, 1,81,953 ha (59%) in 2009, and 1,68,476 ha (54%) in 2019. In contrast, agricultural lands increased at an alarming rate in the study area as it increased from 21,787 ha (7%) in 1988 to 34,018 ha (11%) in 1999, 49,229 ha (16%) in 2009, and, 78,704 ha (25%) in 2019 which became a second aerial position in the study area. Shrub land occupies the third aerial position in the study area which substantially increased from 40,387 ha (13%) in 1988 to 63,315 ha (20%) in 1999 and slightly increased between 1999 and 2009, but tremendously decreased to 44,901 ha (14%) in 2019.

**Table 3. Accuracy assessment of LULC classes for 1988, 1999, 2009 and 2019.**

| Year | LULC classes | Producer's accuracy | User's accuracy | Overall accuracy | Kappa |
|------|--------------|---------------------|-----------------|------------------|-------|
|      | Forest       | 81%                 | 100%            |                  |       |
|      | Agriculture  | 97%                 | 97%             |                  |       |
| 1988 | Shrub        | 100%                | 70%             | 92%              | 90%   |
|      | Settlement   | 88%                 | 97%             |                  |       |
|      | Waterbody    | 100%                | 97%             |                  |       |
|      | Forest       | 83%                 | 100%            |                  |       |
|      | Agriculture  | 91%                 | 97%             |                  |       |
| 1999 | Shrub        | 97%                 | 97%             | 93%              | 91%   |
|      | Settlement   | 100%                | 83%             |                  |       |
|      | Waterbody    | 96%                 | 87%             |                  |       |
|      | Forest       | 83%                 | 97%             |                  |       |
|      | Agriculture  | 97%                 | 93%             |                  |       |
| 2009 | Shrub        | 94%                 | 97%             | 94%              | 93%   |
|      | Settlement   | 100%                | 90%             |                  |       |
|      | Waterbody    | 100%                | 93%             |                  |       |
|      | Forest       | 83%                 | 97%             |                  |       |
|      | Agriculture  | 91%                 | 100%            |                  |       |
| 2019 | Shrub        | 100%                | 93%             | 94%              | 93%   |
|      | Settlement   | 100%                | 90%             |                  |       |
|      | Waterbody    | 100%                | 90%             |                  |       |

Settlement area gradually increased during the study period i.e., the area increased from 7367 ha (2%) in 1988 to 9479 ha (3%) in 1999, 10127 ha (3%) in 2009, and 13019 ha (4%) in 2019. The area of water bodies gradually decreased from 6,142 ha (2%) in 1988 to 5652 ha (2%) in 1999, 5533 ha (2%) in 2009, and 5217 ha (2%) in 2019.

Fig 6 shows the relative changes in the area of different LULC types during the period of 1988–1999, 1999–2009, 2009–2019, and 1988–2019. The change in forest area and shrubland was found very high during 1988–1999 when compared with the change between 1999–2000 and 2009–2019. In contrast, the areal change in agriculture and settlement land use classes was found higher during 2009–2019 than the change observed during 1988–1999 and 1999–2009 (S1 Table). The change in water body was found higher during 1988–1999 than in the other two periods 1999–2009 and 2009–2019.

Throughout the study period, there was a negative change or continuous reduction in forest lands with an average annual rate of 2,134.13 ha/year and a positive change or continuous increase in agricultural land and settlement area with an average annual rate of 1,836.03 ha/year and 182.32 ha/year, respectively, which might be due to more upland cultivation as well as paddy in the lowland areas caused by the increasing population. It is noted that the stability of the hills has been reducing and several landslides/mudflows occurred in the recent past.

Shrub lands showed a positive change in all three decadal periods with an overall increased rate of 145.61 ha/year. Waterbody was found continuously decreasing during the whole study period at an average annual rate of 29.84 ha/year which indicates the drying of waterbody in the study area due to the reduction of upstream flow and withdrawal of water for riverside cultivation practices by constructing artificial barrage.

## LULC change patterns

The patterns of LULC classes over the period of 1988–1999, 1999–2009, 2009–2019, and 1988–2019 conducted by matrix operation are showed in Figs 6–9. Results of the change matrix between 1988–1999 showed that 53.17% of the

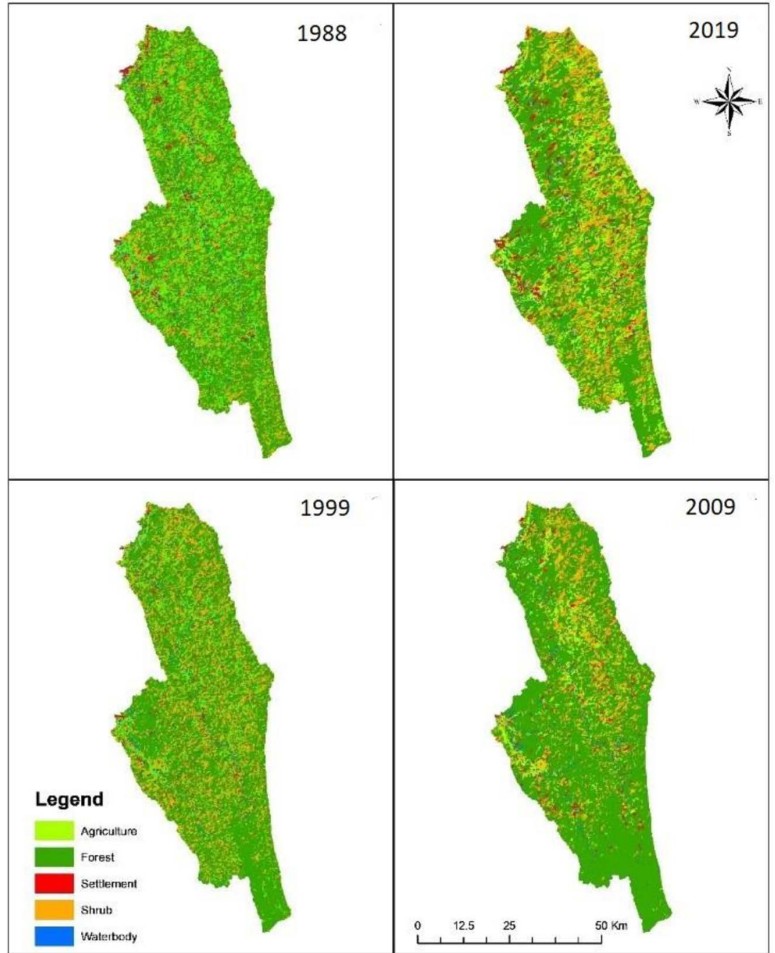

**Fig 4. LULC maps of SMW in 1988, 1999 2009, and 2019 obtained from image classification.**

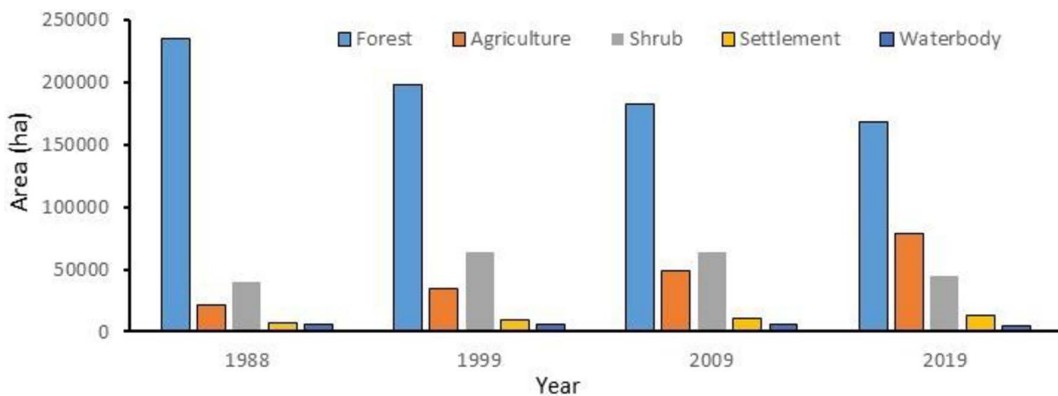

**Fig 5. Temporal change of different LULC in the Sangu-Matamuhuri watershed.**

**Table 4. Areal distribution of LU classes of study area in 1988, 1999, 2009 and 2019.**

| Class name | Year 1988 | | Year 1999 | | Year 2009 | | Year 2019 | |
|---|---|---|---|---|---|---|---|---|
| | Area (ha) | % | Area (ha) | % | Area (ha) | % | Area (ha) | % |
| Forest | 234634 | 76% | 197852 | 64% | 181953 | 59% | 168476 | 54% |
| Agriculture | 21787 | 7% | 34018 | 11% | 49229 | 16% | 78704 | 25% |
| Shrub | 40387 | 13% | 63315 | 20% | 63476 | 20% | 44901 | 14% |
| Settlement | 7367 | 2% | 9479 | 3% | 10127 | 3% | 13019 | 4% |
| Waterbody | 6142 | 2% | 5652 | 2% | 5533 | 2% | 5217 | 2% |
| Total | 310317 | 100% | 3,10,317 | 100% | 310317 | 100% | 310317 | 100% |

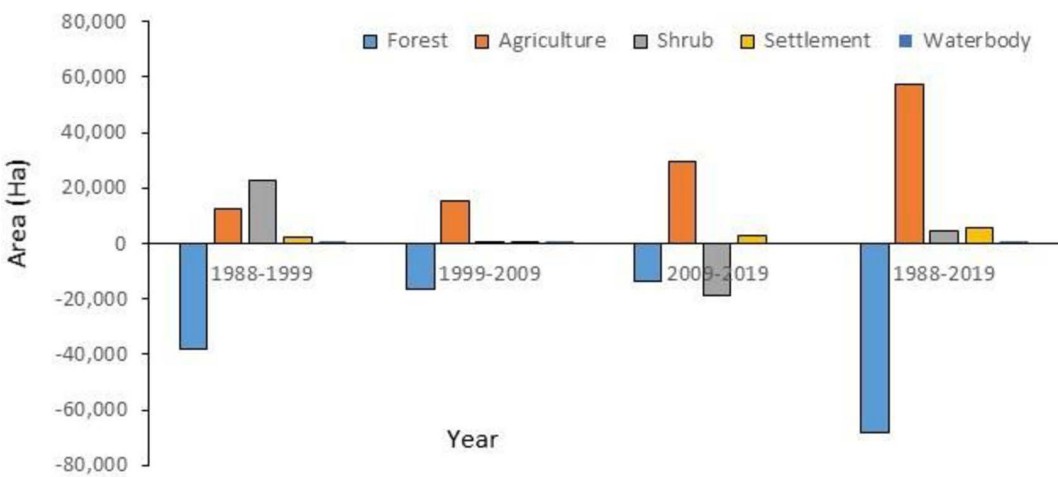

**Fig 6. The relative change in LULC in Sangu-Matamuhuri watershed.**

forest land was retained as the same land use (forest) in 1999, whereas the remaining 22.43% were converted into other land use classes predominantly shrub (14.28%) and agriculture (6.17%) (Fig 7).

During the studied period some of the agricultural areas were converted into other land use classes, but much of the agricultural area was gained from the conversion of forest land. The area retained under shrubland was 3.85% in 1999, but much of its area was converted mainly into forest and agricultural area and again gained much from the conversion of the same land use classes. In this period, there was also a considerable increase in the area of settlement mainly because of the conversion of forest, agriculture, and shrubland into this category (S2 Table). It was observed that some waterbody was converted into forest which might be due to classification error and some waterbodies were gained from the conversion of forest land resulting from soil erosion.

The result of the change matrix between 1999 and 2009 showed that a total of 44.11% of forest area were retained as forests in 2009 and the remaining 19.66% was converted into other LU classes mainly to shrubland and agriculture (Fig 8). Out of 10.96% agricultural area in 1999, much of the area (7.3%) was converted into other LU classes but gained again significant portion from the conversion of forest land in 2009 (S3 Table). During this period the settlement area increased because of the conversion of the forest, agriculture, and shrubland whereas the area of water body decreased because of conversion into other land use classes.

During the period 2009–2019, the considerable change in land use was the conversion of forest land into shrub land corresponding to 7.87%, and agricultural land corresponding to 11.66% of the total area (Fig 9). Much of the agricultural area was found converted into other land classes in 2019 but it was increased in 2019 by gaining mostly from forest and

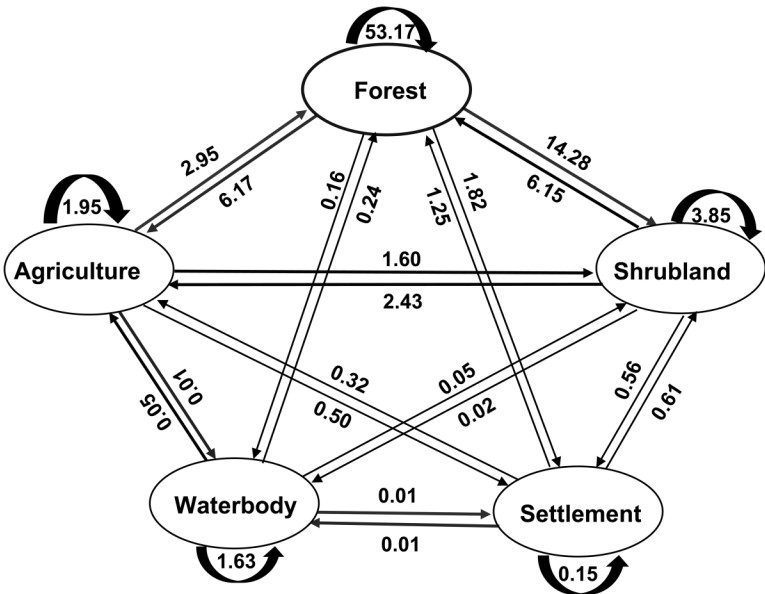

**Fig 7. Change matrix of LULC (%) in SMW between 1988 and 1999.**

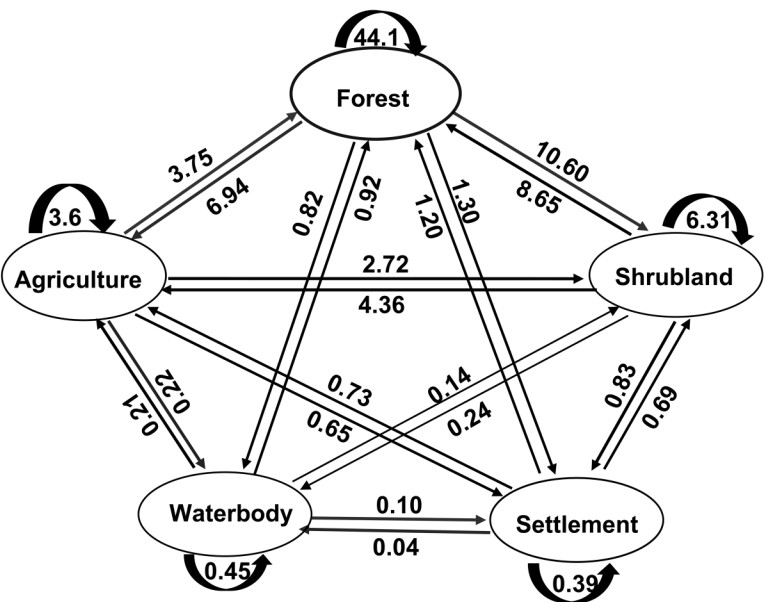

**Fig 8. Change matrix of LULC (%) in SMW between 1999 and 2009.**

shrub land use classes. About 11.12% of shrubland was converted into other land use classes and gained 17.11% from the other land use classes in 2019. The settlement area was increased by gaining mostly from the forest and agricultural land in 2019. The area of waterbody (including small creeks, river side and raised sandy land in the river) was found converted into agriculture and forest land and thus decreased from 1.78% in 2009 to 1.68% in 2019 (S4 Table).

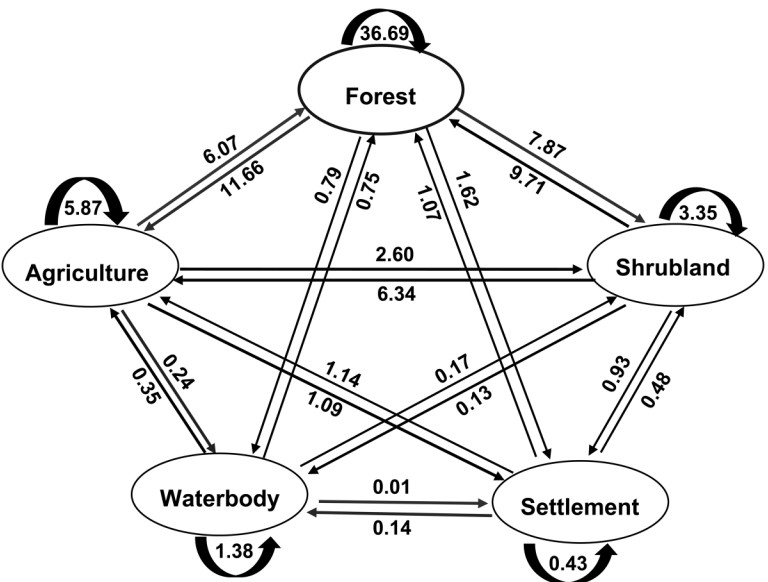

**Fig 9. Change matrix of LULC (%) in SMW between 2009 and 2019.**

Overall during the last 31 years (1988–2019), the most significant change in LULC category was the conversion of forest land to agricultural land corresponding to 17.14% of the total area (Fig 10 and S5 Table). The conversion of forest land to shrub land was also notable (10.99%). On the contrary, the area under agriculture was significantly increased mainly because of the conversion of forest land to this category. The shrub land and settlement also increased mainly due to the conversion of forest land to this category. Some area of the waterbody (including small creeks, river side and raised sandy land in the river) was found converted into other LULC category.

## Overall gain and loss

The gain and loss percentages of the total area of LULC in the SMW during the study period are shown in Table 5. During this period 1988–2019, the highest land loss was observed for the forest class (31.21%) whereas the highest gain (22.77%) was achieved in agricultural areas. The increase in shrub land with the exception during 2009–2019 and settlement area were also observed but the decrease in water bodies was noticeable. However, there were regular periodical changes in other land use categories observed during 1988–1999, 1999–2009, and 2009–2019.

## Discussion

The result of the study showed that there was substantial shrinkage of forest area in the studied watershed with an annual loss of 2,134.13 ha/year. Loss of forest land or even conversion into shrub or upland planted fields was also found as the mid-step to convert the forest to agriculture. Beyond the forest area reduction, there was a continuous expansion of agricultural area with the rate of 1,836.03 ha/year through conversion or sacrifice of forest land. A similar trend was observed in a study carried out for whole CHTs region by Roy et al. [35] where 3,06,200 ha of hill forest area was found to decrease during the period of 1989–2014 at the rate of 12,248 ha per year, 2,99,082 ha of shrub land and 25,874 ha of cropland also increased at the rate of 11,963 ha/year and 1,035 ha/year respectively. Four groups of actors were reported to be responsible for deforestation in this region (i) the native forest dwellers with their high population growth, (ii) migrants, who move to the forests, (iii) the timber industries that cut down too many trees, and (iv) the government policies that regulate

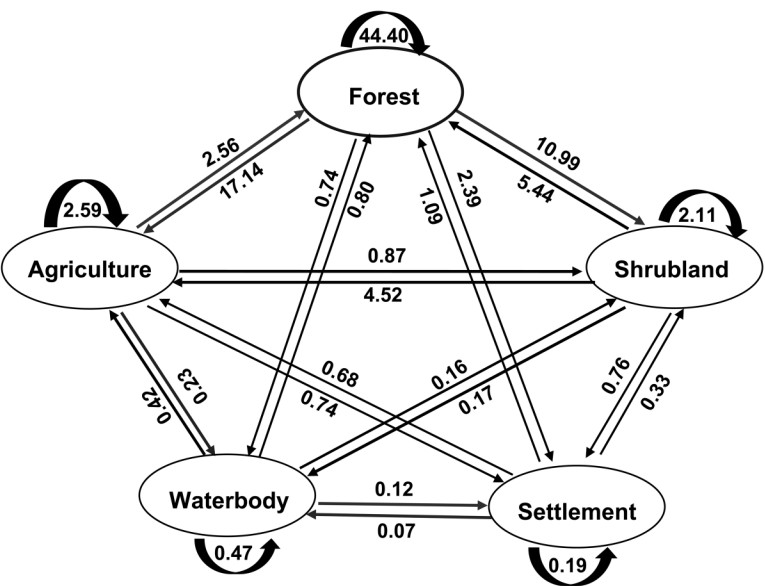

**Fig 10. Change matrix of LULC (%) in SMW between 1988 and 2019.**

**Table 5. Gain and loss percentage of total area in the SMW area during 1988-1999, 1999-2009, 2009-2019 and 1988-2019 intervals.**

| LULC Classes | 1988 - 1999 | | 1999 - 2009 | | 2009 - 2019 | | 1988 - 2019 | |
|---|---|---|---|---|---|---|---|---|
| | Loss (%) | Gain (%) | Loss (%) | Gain (%) | Loss (%) | Gain (%) | Loss (%) | Gain (%) |
| **Forest** | 22.44 | 0.11 | 19.65 | 14.52 | 21.94 | 1.76 | 31.21 | 0.10 |
| **Agriculture** | 5.07 | 9.01 | 7.36 | 12.26 | 9.99 | 19.49 | 4.43 | 22.77 |
| **Shrub** | 9.16 | 16.55 | 14.09 | 14.15 | 17.11 | 11.12 | 10.90 | 12.36 |
| **Settlement** | 2.22 | 2.90 | 2.66 | 2.87 | 2.83 | 3.77 | 2.18 | 4.01 |
| **Waterbody** | 0.35 | 0.19 | 1.37 | 1.33 | 1.40 | 1.30 | 1.50 | 1.21 |

tree cutting and prevent illegal cutting [66]. Deforestation was also accelerated by a disregard for the rights to traditional forest use and management. 49% of the households in CHTs depend on agriculture for their livelihood [66] compared to 46% in rural Bangladesh [43,67]. Besides, plain land agriculture, swidden agriculture known as 'Jhum' has been widely practiced by communities living in the hills. Bandarban district has the largest proportion of jhum land (57%) than other hill districts. The jhum cycle once ranged from 12 to 15 years was now shortened to 2–3 years seriously affecting soil quality and yield. Utilizing the steeply sloping lands for agriculture without any management practices also makes the land vulnerable to degradation. Pineapple cultivation on hill slopes is a faulty agricultural practice that causes severe soil erosion [68]. However, this LULC change has a large negative impact upon watershed ecosystem services such as water regulation, soil conservation, and carbon sequestration. Deforestation disrupts the natural water cycle, leading to increased surface runoff, soil erosion, and reduced water availability; exposes the soil, making it vulnerable to erosion and degradation; releases the stored carbon back into the atmosphere, contributing to climate change; reduces the capacity of forests to absorb carbon dioxide in the future and impacting the overall health and resilience of the ecosystem [34,58,69,70].

Settlement in the study area was found gradually increased from 7,367 ha (2%) in 1,988 to 9,479 ha (3%) in 1999, 10,127 ha (3%) in 2009, and 13,019 ha (4%) in 2019. GoB and FAO [56] in a joint technical study reported that during 2001–2011 the CHTs region experienced massive population growth (19.5%). Accordingly, this study area also had

the highest population growth. As a result, land use in the study area was under additional pressure to find housing for individuals.

Another alarming scenario was the gradual decline of water bodies that serves as the primary sources of drinking water for the local indigenous community and also act as the sources of irrigation water during the dry seasons. The result of this study showed that water body was decreased from 6,142 ha in 1988 to 5,652 ha in 1999, 5,533 ha in 2009, and 5,217 ha in 2019 which indicate drying of water body (including natural lake, water springs, small creeks and canals of Sangu and Matamuhuri Rivers) in the study area. Roy et al. [18] showed that the river area in CHTs increased by 1,209 ha from 1989 to 2000, but decreased considerably to 1,509 ha in 2014 while other inland water bodies increased by 77 ha from 1989 to 2000 but decreased to 160.22 ha during the period 2000–2014. Along with this, river flow is certainly reduced except in monsoon season when it overflows and the sub-surface area goes underwater as observed during field visits in the study period. This situation makes the local communities vulnerable to natural disasters, such as flash floods, landslides in one side and droughts on the other. It was reported that people have to travel long distances to fetch drinking water during dry season [66]. Their livelihoods including agriculture, forestry, fishing are severely affected which leads them towards economic hardship and social instability [34,38,66]. Proper upland water management including community led site specific soil and water conservation works such as reforestation measures with native tree species of high water retention capacity, composting, mulching, intercropping, making percolation pond, constructing mini-check dam, etc. should be ensured to overcome this adverse situation in the study area.

Change matrices generated from the four-study period revealed that between the years 1988 and 1999, forest area contributed 6.17% (19,155 ha) to the agricultural area, and 14.28% (44,323 ha) to shrubland. During the period of 1999–2009 the contribution of forest area increased by 6.94% (21,518 ha) to the agricultural area but decreased to 10.60% (32,880 ha) in the case of shrub land. Subsequently, between the years 2009 and 2019, forest area contributed 11.66% (36,180 ha) to the agricultural area and 7.87% (24,487 ha) to shrub land. The total conversion from the forest to agricultural area and shrub land over the last 31 years was 17.14% (53,210 ha) and 10.99% (33,923 ha), respectively. On the contrary, forest land undertook 2.56% (7,942 ha) area from agriculture and 5.44% (16,890 ha) from shrub land during 1988–2019. A vast area of forest land was found converted into agriculture and shrub land in the form of shifting cultivation, mixed orchard, cash crop cultivation, bushy plants, etc. Due to this deforestation and agricultural expansion, soil erosion, and quick runoff during rains occurs which results in flash floods and landslides sometimes [34,58,70]. The area under the settlement category was significantly increased resulting mainly from forest land which was 1.82% (5663 ha) during 1988–1999, 1.30% (40224 ha) during 1999–2009, and 1.62% (5030 ha) during 2009–2019. Agriculture and shrubland also contributed to the expansion of settlement area. About 925 ha of water body was found reduced during the period of 1988–2019 with conversion mainly from forest land and agriculture.

The Gain and loss scenario for different LULC classes over three decadal periods in SMW area indicated that the highest loss of forest area occurred during the period of 1988–1999 in comparison with 1999–2009 and 2009–2019. The reason might be the large scale removal of trees by the people living inside the forest area during this period who were basically suspicious about their right due to peace accord that was signed in 1997 between The Government of Bangladesh and tribal organization with an objective to elevate political, social, cultural, educational and financial rights and to expedite socio-economic development process of all citizens in CHTs [56]. The highest gain in agriculture and shrubland was found during 2009–2019 in comparison with 1988–1999 and 1999–2009 which might be due to more upland cultivation as well as paddy in the lowland areas caused by the increasing population inside the forest areas during this period. Overpopulation and expansion of unplanned agriculture in the study area causes gradual deforestation which ultimately resulting biodiversity loss, soil degradation and significant contribution to global climate change, etc. [42,43]. Continuation of this scenario will result in more degraded land in the near future. Hence, considering the social, economic, and environmental aspects of the studied watershed an integrated watershed management plan including sustainable land management practices such as reforestation measures, site specific soil and water conservation works, engaging local

communities in the planning and implementation of those management practices along with proper policy and regulatory framework should be developed and implemented.

During LULC change analysis, degraded forests were found in several areas reflecting as shrubland or sometimes as agricultural land. For resolving this confusion, rigorous field visits were conducted throughout the study area, and found upland cultivation increased to a greater extent. Almost all hilly areas were found encroached and ultimately faced this problem. Mapping of water bodies were found difficult due to the shallow water and dense canopy along the river. Difficulty was further increased with 30 m spatial resolution. These problems were minimized by collecting ground truth data. This study demonstrates the significance of integrating Remote Sensing and GIS for change detection study of land use and land cover of an area as it provides crucial information about the spatial distribution as well as nature of land cover changes. Over 92% accuracy results of Land use and land cover map reveals that combining visual interpretation with supervised classification of satellite imagery is a useful technique for the documentation of changes in land use and cover of an area.

## Conclusion

This study assessed and monitored the changing pattern of LULC in Sangu-Matamuhuri watershed during 1988, 1999, 2009, and 2019. Five LULC classes with the highest accuracy level present valuable information about loss of forest land, agricultural expansion, urbanization and water resource situation in the upper part of Sangu-Matamuhuri watershed. Out of all five land use classes, the major changes occurred in the forest area which showed a drastic decrease, and at the same time agriculture, shrubland, and settlement areas showed a continuous increase during the study period. A vast area of forest land was transformed into agriculture in the form of shifting cultivation, mixed orchard, cash crop cultivation, etc. caused soil erosion, and quick runoff during rains resulting in flash floods and landslides. On the contrary, the area under agriculture was substantially increased mainly because of the conversion of forest land to this category. The shrub land and settlement also increased mainly due to the conversion of forest land to this category. But the area of water bodies, the only source of drinking water for the indigenous community and irrigation water in the dry season, was found decreased which indicates the drying of water bodies in the study area. The observed changing trend of LULC in the study area posed a serious threat to Sangu-Matamuhuri watershed resources and thereby its ecosystem services. As a result, there would be more water shortage in the near future during the dry season and excess water in the rainy season causing prolonged drought, flash flood, land slide, declining soil productivity, deteriorating water quality which ultimately affect both low land and upland communities in the watershed.

Hence, in order to mitigate the negative effects of LULC changes, proper management of watershed resources is required. An integrated watershed management program should be undertaken considering all biophysical, socioeconomic, environmental and institutional issues beyond as well within the area. Identifying degraded micro-watershed, adequate forest restoration measures including enrichment planting, assisted natural regeneration, site specific soil and water conservation works, organizing stakeholder consultation meeting with all institutions and leaders and training of community people on resilient livelihood skill development should be carried out immediately in the watershed area.

## Supporting information

**S1 Table. The relative change in LULC types (showing % change in bracket) in the study area.**
(TIFF)

**S2 Table. Transition matrix of LULC change (ha) between 1988 and 1999.**
(TIFF)

**S3 Table. Transition matrix of LULC change (ha) between 1999 and 2009.**
(TIFF)

**S4 Table. Transition matrix of LULC change between 2009 and 2019.**
(TIFF)

**S5 Table. Transition matrix of LULC change in SMW between 1988 and 2019.**
(TIFF)

## Acknowledgments

Authors are thankful to the Bangladesh Forest Department, Ministry of Environment, Forests and Climate Change; Bangladesh Water Development Board, and Ministry of Water Resources, for their all-out support to carry out field studies.

## Author contributions

**Conceptualization:** Ajit Kumar Rudra, A.K.M. Rashidul Alam.

**Data curation:** Ajit Kumar Rudra, A.K.M. Rashidul Alam.

**Formal analysis:** Ajit Kumar Rudra.

**Investigation:** Ajit Kumar Rudra, A.K.M. Rashidul Alam.

**Methodology:** Ajit Kumar Rudra, A.K.M. Rashidul Alam.

**Resources:** Ajit Kumar Rudra, A.K.M. Rashidul Alam.

**Software:** Ajit Kumar Rudra.

**Supervision:** A.K.M. Rashidul Alam.

**Validation:** A.K.M. Rashidul Alam.

**Writing – original draft:** Ajit Kumar Rudra.

**Writing – review & editing:** Ajit Kumar Rudra, A.K.M. Rashidul Alam.

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
