## [Decision Letter · Decision Letter 0]

Dear Dr. Alam,

Thank you for submitting your manuscript to PLOS ONE. After careful consideration, we feel that it has merit but does not fully meet PLOS ONE’s publication criteria as it currently stands. Therefore, we invite you to submit a revised version of the manuscript that addresses the points raised during the review process.

We look forward to receiving your revised manuscript.

Kind regards,

Mitiku Badasa Moisa

Academic Editor

PLOS ONE

Journal Requirements:

2. We note that Figures 1 and 4 in your submission contain [map/satellite] images which may be copyrighted. All PLOS content is published under the Creative Commons Attribution License (CC BY 4.0), which means that the manuscript, images, and Supporting Information files will be freely available online, and any third party is permitted to access, download, copy, distribute, and use these materials in any way, even commercially, with proper attribution. For these reasons, we cannot publish previously copyrighted maps or satellite images created using proprietary data, such as Google software (Google Maps, Street View, and Earth). For more information, see our copyright guidelines: http://journals.plos.org/plosone/s/licenses-and-copyright.

a. You may seek permission from the original copyright holder of Figures 1 and 4 to publish the content specifically under the CC BY 4.0 license.  

3. We note you have included a table to which you do not refer in the text of your manuscript. Please ensure that you refer to Table 2 in your text; if accepted, production will need this reference to link the reader to the Table.

Additional Editor Comments:

Your manuscript entitled "Spatiotemporal changing pattern of land use and land cover of Sangu-Matamuhuri watershed in the South-Eastern part of Bangladesh", which you submitted to Cogent Food & Agriculture, has now been reviewed.

The reviews, included at the bottom of the letter, indicate that your manuscript could be suitable for publication following revision. We hope that you will consider these suggestions, and revise your manuscript.

Reviewers' comments:

Reviewer's Responses to Questions

**Comments to the Author**

1. Is the manuscript technically sound, and do the data support the conclusions?

Reviewer #1: Yes

Reviewer #2: Yes

2. Has the statistical analysis been performed appropriately and rigorously?

Reviewer #1: Yes

Reviewer #2: Yes

3. Have the authors made all data underlying the findings in their manuscript fully available?

Reviewer #1: Yes

Reviewer #2: No

4. Is the manuscript presented in an intelligible fashion and written in standard English?

Reviewer #1: Yes

Reviewer #2: Yes

Reviewer #1: The manuscript needs some conceptual and factual corrections and clarifications. The details attached in Microsoft Word document using track changes functionality. Therefore, the authors should incorporate the comments and suggestions attached.

Reviewer #2: I appreciate the opportunity to review your manuscript titled Spatiotemporal Changing Pattern of Land Use and Land Cover of Sangu-Matamuhuri Watershed in the Southeastern Part of Bangladesh. The study provides valuable insights into the dynamics of land use and land cover (LULC) changes over a significant period, highlighting critical environmental and socio-economic implications.

Overall Assessment:

The manuscript is well-structured, with a clear research question and robust methodology. The use of multi-temporal satellite imagery and thorough accuracy assessments supports the findings effectively. However, there are several areas where enhancements can be made to improve clarity, depth, and impact.

Major Comments:

Summary of Key Findings:

While the conclusion summarizes the key findings effectively, I recommend expanding the discussion to link these findings to broader implications, such as climate change and biodiversity loss. This will emphasize the urgency of the observed changes.

Ecosystem Services:

The manuscript mentions threats to ecosystem services but could elaborate on how LULC changes impact services like water regulation and soil conservation. This connection would provide a more comprehensive understanding of the ecological consequences.

Recommendations for Future Actions:

I encourage you to provide specific, actionable recommendations for reforestation and sustainable land management practices. Including examples such as agroforestry techniques or alternative livelihoods for forest-dependent communities would enhance the practicality of your suggestions.

Water Resources and Community Impacts:

Expand the discussion on the socio-economic impacts of water scarcity, particularly how it affects agricultural productivity and local communities. Addressing potential strategies for water conservation would also be beneficial.

Urgency and Long-Term Outlook:

Strengthening the sense of urgency regarding LULC changes will underscore the importance of immediate intervention. Discussing long-term risks associated with continued degradation can help frame the necessity for comprehensive land-use planning.

Minor Comments:

Clarity and Flow: Improve transitions between points in the conclusion to create a cohesive narrative.

Consistency in Data Presentation: Reduce redundancy by summarizing key findings without repeating exact figures from previous sections.

Technical Terminology: Avoid abbreviations without explanation to ensure accessibility for all readers.

Typographical and Grammatical Issues: Proofread the manuscript for grammatical errors and awkward phrasing.

Ethical Considerations:

I found no ethical concerns regarding research methodology or data collection. The manuscript does not raise issues of dual publication or conflict of interest; if applicable, please explicitly state this.

Conclusion:

The manuscript presents important findings on LULC changes in the Sangu-Matamuhuri Watershed. With minor revisions addressing the areas noted above, it will significantly contribute to the fields of environmental studies, geography, and land-use management. I recommend minor revisions before acceptance for publication. Thank you for the opportunity to review this important work.

**Do you want your identity to be public for this peer review?** For information about this choice, including consent withdrawal, please see our Privacy Policy

Reviewer #1: **Yes: ** Wondafrash Genet

Reviewer #2: No

---

## [Author Response · Author response to Decision Letter 1]

30 Dec 2024

Responses of the authors to the comments of the Editors and Reviewers

Response to editor’s comments:

Authors: PLOS One Manuscript Body Formatting Guidelines and Title, Authors, Affiliations Formatting Guidelines have been followed accordingly during manuscript preparation.

2. We note that Figures 1 and 4 in your submission contain [map/satellite] images which may be copyrighted. All PLOS content is published under the Creative Commons Attribution License (CC BY 4.0), which means that the manuscript, images, and Supporting Information files will be freely available online, and any third party is permitted to access, download, copy, distribute, and use these materials in any way, even commercially, with proper attribution. For these reasons, we cannot publish previously copyrighted maps or satellite images created using proprietary data, such as Google software (Google Maps, Street View, and Earth). For more information, see our copyright guidelines: http://journals.plos.org/plosone/s/licenses-and-copyright.

a. You may seek permission from the original copyright holder of Figures 1 and 4 to publish the content specifically under the CC BY 4.0 license.

Authors: Figure 1 in our submission is basically prepared by authors in earlier publication (Title: ‘Streamflow characteristics of Sangu-Matamuhuri watershed in the Southeastern part of Bangladesh’, Published in Heliyon journal, Volume 9, Issue 3, March 2023, e14559, Link: https://www.sciencedirect.com/science/article/pii/S2405844023017668. https://www.cell.com/heliyon/pdf/S2405-8440(23)01766-8.pdf).

Figure 4 in our submission is also prepared by authors. But when submitted earlier for publication in other Journal, it was unauthorizely published as a preprint by Research Gate, Research Square.

3. We note you have included a table to which you do not refer in the text of your manuscript. Please ensure that you refer to Table 2 in your text; if accepted, production will need this reference to link the reader to the Table.

Authors: Table 2. LULC classification scheme used for the study was modified from ADB, 2001. It was already mentioned in the Text (Page 11, Line 224, 230) and reference (Page 32, Line 682) in accordance with guideline. Please see page 11, line 216, 221.

Additional Editor Comments:

Your manuscript entitled "Spatiotemporal changing pattern of land use and land cover of Sangu-Matamuhuri watershed in the South-Eastern part of Bangladesh", which you submitted to Cogent Food & Agriculture, has now been reviewed.

Authors: The Author didn’t submit manuscript titled ‘"Spatiotemporal changing pattern of land use and land cover of Sangu-Matamuhuri watershed in the South-Eastern part of Bangladesh" to ‘Cogent Food & Agriculture’.

Responses to Reviewers comments:

Reviewer #1: The manuscript needs some conceptual and factual corrections and clarifications. The details attached in Microsoft Word document using track changes functionality. Therefore, the authors should incorporate the comments and suggestions attached.

Authors: All conceptual and factual corrections and clarifications as mentioned in Microsoft Word document of the manuscript using track changes functionality are accomplished and the comments and suggestions attached therein are also incorporated in the manuscript.

A marked-up copy of our manuscript that highlights changes made to the original version is uploaded as a separate file labeled 'Revised Manuscript with Track Changes'.

Major Comments:

Summary of Key Findings:

While the conclusion summarizes the key findings effectively, I recommend expanding the discussion to link these findings to broader implications, such as climate change and biodiversity loss. This will emphasize the urgency of the observed changes.

Authors: The key findings of the study are elaborately described in conclusion part with actionable steps or recommendations for broader implications.

Ecosystem Services:

The manuscript mentions threats to ecosystem services but could elaborate on how LULC changes impact services like water regulation and soil conservation. This connection would provide a more comprehensive understanding of the ecological consequences.

Authors: According to editor’s advice, impact of LULC changes on ecosystem services including water regulation and soil conservation are elaborated for better understanding of the ecological consequences in the discussion chapter.

Recommendations for Future Actions:

I encourage you to provide specific, actionable recommendations for reforestation and sustainable land management practices. Including examples such as agroforestry techniques or alternative livelihoods for forest-dependent communities would enhance the practicality of your suggestions.

Authors: Specific and actionable recommendations for reforestation and sustainable land management practices including examples such as agroforestry techniques or alternative livelihoods for forest-dependent communities are provided in the conclusion chapter.

Water Resources and Community Impacts:

Expand the discussion on the socio-economic impacts of water scarcity, particularly how it affects agricultural productivity and local communities. Addressing potential strategies for water conservation would also be beneficial.

Authors: Socio-economic impacts of water scarcity with particular effect upon agricultural productivity and local communities are elaborately discussed in the discussion chapter. Potential strategies for water conservation are also addressed.

Urgency and Long-Term Outlook:

Strengthening the sense of urgency regarding LULC changes will underscore the importance of immediate intervention. Discussing long-term risks associated with continued degradation can help frame the necessity for comprehensive land-use planning.

Authors: Short and long-term strategies in order to combat risks associated with continued degradation due to LULC changes are addressed for comprehensive land-use planning.

Minor Comments:

Clarity and Flow: Improve transitions between points in the conclusion to create a cohesive narrative.

Authors: Transitions between points in the conclusion chapter are improved.

Consistency in Data Presentation: Reduce redundancy by summarizing key findings without repeating exact figures from previous sections.

Authors: The key findings of the study are elaborately described for broader implications. Repetitions of exact figures from previous sections are avoided.

Technical Terminology: Avoid abbreviations without explanation to ensure accessibility for all readers.

Author: Abbreviations used throughout this study are explained accordingly.

Typographical and Grammatical Issues: Proofread the manuscript for grammatical errors and awkward phrasing.

Authors: Proof reading has been performed by native speaker and conducted necessary corrections.

Ethical Considerations:

I found no ethical concerns regarding research methodology or data collection. The manuscript does not raise issues of dual publication or conflict of interest; if applicable, please explicitly state this.

Author: This article titled “Spatiotemporal pattern of land use and land cover of upper Sangu-Matamuhuri watershed in the South-Eastern Bangladesh” is an original research performed by authors. All ethical practices have been followed in the writing of this manuscript.

---

## [Decision Letter · Decision Letter 1]

Dear Dr. Alam, 

We look forward to receiving your revised manuscript.

Kind regards,

Dessalegn Obsi Gemeda, PhD

Academic Editor

PLOS ONE

Journal Requirements:

Additional Editor Comments:

Dear authors,

Thank you for submitting your work to PLOS One.

Based on the reviwer comments and my own review, this paper can be suitable for pbilcation if you address the following points before my recommendation. You can address and re-submit within maximum of 10 days.

While the conclusion summarizes the key findings effectively, expanding the discussion to link these findings to broader implications, such as climate change and biodiversity loss are required from the authors.

The manuscript mentions threats to ecosystem services but could elaborate on how LULC changes impact services like water regulation and soil conservation is remains unclear.

Expand the discussion on the socio-economic impacts of water scarcity, particularly how it affects agricultural productivity and local communities. Addressing potential strategies for water conservation would also be beneficial.

Discussing long-term risks associated with continued degradation can help frame the necessity for comprehensive land-use planning.

Clarity and Flow: Improve transitions between points in the conclusion to create a cohesive narrative.

Consistency in Data Presentation: Reduce redundancy by summarizing key findings without repeating exact figures from previous sections.

Technical Terminology: Avoid abbreviations without explanation to ensure accessibility for all readers.

Typographical and Grammatical Issues: Proofread the manuscript for grammatical errors and awkward phrasing.

The present study is at a watershed level. Hence, your conclusion should reflect the implications of the land use change on watershed management. So, you should add a paragraph or two in this regard.

Forest degradation refers to the decline in forest quality, health, and functionality due to various factors. In your study you focused on land use change. So, change this to ‘loss of forest land’.

Reviewers' comments:

Reviewer's Responses to Questions

**Comments to the Author**

Reviewer #1: All comments have been addressed

2. Is the manuscript technically sound, and do the data support the conclusions?

Reviewer #1: Yes

3. Has the statistical analysis been performed appropriately and rigorously?

Reviewer #1: Yes

4. Have the authors made all data underlying the findings in their manuscript fully available?

Reviewer #1: (No Response)

5. Is the manuscript presented in an intelligible fashion and written in standard English?

Reviewer #1: Yes

Reviewer #1: The authors of this article have included all the comments and suggestions, which I raised in the early phase of reviewing the manuscript. The only issue I have is that they declared that they did not get fund for conducting the project, but they eventually acknowledged Bangladesh Forest Department. What kind of support did they get from the institution?

**Do you want your identity to be public for this peer review?** For information about this choice, including consent withdrawal, please see our Privacy Policy

Reviewer #1: **Yes: ** Wondafrash G. Degu

---

## [Author Response · Author response to Decision Letter 2]

7 Apr 2025

Responses of the authors to the comments of the Academic Editor and Reviewers

Response to editor’s comments:

Journal Requirements:

Authors: According to PLOS One submission guidelines, reference list is reviewed and changes made to the reference list according to citation in the text are mentioned as follows:

Reference numbers are corrected according to citation in the text:

- Reference number 21-27 were wrongly numbered in the text and so corrected as 6-12 according to text. - Reference number 13 is newly included. - Reference number 28, 29 are replaced as 14 and 15 according to text. - Reference number 7 is replaced as 16 according to text. - Reference number 31-37 are replaced as 17-23 respectively according to text. - Reference number 6 is replaced as 24 according to text. - Reference number 8-20 are replaced as 25-37 according to text. - Reference number 30, 40, 43 are subtracted due to their duplication. - Reference number 41-42 are replaced as 40-41 according to text. - Reference number 42-43 are newly included according to text. - Reference number 66 is subtracted due to its non-relevancy in the text. - Reference number 67 is replaced as 66 according to text. - Reference number 68, 69 are replaced as 43, 67 respectively according to text. - Reference number 68, 69 and 70 are newly included. -

Additional Editor Comments:

2. While the conclusion summarizes the key findings effectively, expanding the discussion to link these findings to broader implications, such as climate change and biodiversity loss are required from the authors.

Authors: In the discussion section (Line 419-425, 441-449, 479-487), all key findings are broadly discussed with their implications including climate change and biodiversity loss.

3. The manuscript mentions threats to ecosystem services but could elaborate on how LULC changes impact services like water regulation and soil conservation is remains unclear.

Authors: Impact of LULC changes upon ecosystem services including water regulation and soil conservation are elaborately discussed in the discussion section (Line 419 – 425) as mentioned below:

‘However, this LULC change has a large negative impact upon watershed ecosystem services such as water regulation, soil conservation, and carbon sequestration. Deforestation disrupts the natural water cycle, leading to increased surface runoff, soil erosion, and reduced water availability; exposes the soil, making it vulnerable to erosion and degradation; releases the stored carbon back into the atmosphere, contributing to climate change; and also reduces the capacity of forests to absorb carbon dioxide in the future [34, 58, 69, 70].’

4. Expand the discussion on the socio-economic impacts of water scarcity, particularly how it affects agricultural productivity and local communities. Addressing potential strategies for water conservation would also be beneficial.

Authors: Socio-economic impacts of water scarcity relating to agricultural productivity and local communities are broadly discussed in the discussion section (Line 441-445) as mentioned below:

‘This situation makes the local communities vulnerable to natural disasters, such as flash floods, landslides in one side and droughts on the other. It was reported that people have to travel long distances to fetch drinking water during dry season [66]. Their livelihoods including agriculture, forestry, fishing are severely affected which leads them towards economic hardship and social instability [34, 38, 66].’

Potential strategies for water conservation are also addressed (in line 445-449) as follows:

‘Proper upland water management including community led site specific soil and water conservation works such as reforestation measures with native tree species of high water retention capacity, composting, mulching, intercropping, making percolation pond, constructing mini-check dam, etc. should be ensured to overcome this adverse situation in the study area.’

5. Discussing long-term risks associated with continued degradation can help frame the necessity for comprehensive land-use planning.

Authors: In order to frame the necessity for comprehensive land-use planning, long-term risks associated with continued degradation are discussed in the discussion section (Line 479 - 482) as follows:

‘Overpopulation and expansion of unplanned agriculture in the study area causes gradual deforestation which ultimately resulting biodiversity loss, soil degradation and significant contribution to global climate change, etc. [42, 43]. Continuation of this scenario will result in more degraded land in the near future.’

6. Clarity and Flow: Improve transitions between points in the conclusion to create a cohesive narrative.

Authors: In order to create a cohesive narrative, transitions between points in the conclusion are improved.

7. Consistency in Data Presentation: Reduce redundancy by summarizing key findings without repeating exact figures from previous sections.

Authors: Key findings are summarized without repeating exact figures from previous sections.

8. Technical Terminology: Avoid abbreviations without explanation to ensure accessibility for all readers.

Authors: Abbreviations are explained wherever needed.

9. Typographical and Grammatical Issues: Proofread the manuscript for grammatical errors and awkward phrasing.

Authors: Manuscript are proof red for grammatical errors and awkward phrasing.

10. The present study is at a watershed level. Hence, your conclusion should reflect the implications of the land use change on watershed management. So, you should add a paragraph or two in this regard.

Authors: A paragraph regarding implications of the land use change on watershed management is added in the conclusion section (520 – 527) as follows:

‘Hence, in order to mitigate the negative effects of LULC changes, proper management of watershed resources is required. An integrated watershed management program should be undertaken considering all biophysical, socio-economic, environmental and institutional issues beyond as well within the area. Identifying degraded micro-watershed, adequate forest restoration measures including enrichment planting, assisted natural regeneration, site specific soil and water conservation works, organizing stakeholder consultation meeting with all institutions and leaders and training of community people on resilient livelihood skill development should be carried out immediately in the watershed area.’

11. Forest degradation refers to the decline in forest quality, health, and functionality due to various factors. In your study you focused on land use change. So, change this to ‘loss of forest land’.

Authors: The term ‘Forest degradation’ is changed to ‘loss of forest land’.

12. While revising your submission, please upload your figure files to the Preflight Analysis and Conversion Engine (PACE) digital diagnostic tool, https://pacev2.apexcovantage.com/.

Authors: All figure files are uploaded to the Preflight Analysis and Conversion Engine (PACE) digital diagnostic tool, https://pacev2.apexcovantage.com/.

Responses to Reviewers comments:

Reviewer #1: The authors of this article have included all the comments and suggestions, which I raised in the early phase of reviewing the manuscript. The only issue I have is that they declared that they did not get fund for conducting the project, but they eventually acknowledged Bangladesh Forest Department. What kind of support did they get from the institution?

Authors: Yes. Authors did not get any fund from Bangladesh Forest Department. But authors got technical support from Bangladesh Forest Department in terms of performing field studies in the Sangu-Matamuhuri Reserve Forests area which belongs to Bangladesh Forest Department, data collection, etc.

---

## [Editor Report · Decision Letter 2]

Spatiotemporal pattern of land use and land cover changes of upper Sangu-Matamuhuri watershed in the South-Eastern Bangladesh

PONE-D-24-44331R2

Dear Dr. Alam,

We’re pleased to inform you that your manuscript has been judged scientifically suitable for publication and will be formally accepted for publication once it meets all outstanding technical requirements.

Kind regards,

Dessalegn Obsi Gemeda, PhD

Academic Editor

PLOS ONE

Additional Editor Comments (optional):

Dear author,

This is to inform you that the current version is suitable for publication in PLOS ONE Journal.
---

## [Editor Report · Acceptance letter]

PONE-D-24-44331R2

PLOS ONE

Dear Dr. Alam,

I'm pleased to inform you that your manuscript has been deemed suitable for publication in PLOS ONE. Congratulations! Your manuscript is now being handed over to our production team.

Kind regards,

on behalf of

Dr. Dessalegn Obsi Gemeda

Academic Editor

PLOS ONE